# APPLICATION OF SEMI-SUPERVISED GRAPH-BASED LEARNING FOR THE CLASSIFICATION OF SOLUTIONS OF DYNAMICAL SYSTEMS

**Anna Avdyushina**
ITMO University
Saint Petersburg, Russia
avdiushina@itmo.ru

**Elena Avdyushina**
Donetsk State University
Donetsk, Russia
elena.v.a.2023@mail.ru

**Constantin Ruchkin**
construchk@gmail.com

## ABSTRACT

In this article, we consider the application of one of the new methods of semi-supervised graph-based machine learning method (GSSL) for label propagation using the Poisson equation to solve the problem of classifying solutions of dynamic systems by two-dimensional Poincaré section data. The used Poisson learning has advantages over classical Laplace learning since it allows using significantly less labeled data and to spent less time to achieveing the desired accuracy of classification.

We proposed a modification of Poisson learning uses the Nesterov algorithm to find the minimum loss function, to improve the convergence results compared to the classical gradient descent and achieves an accuracy of 92% even with 10 labels per class taking into account accumulation errors and time constraints. This result is acceptable for solving our general task - building an automated system for classifying solutions of dynamic systems in real-time.

The article shows the result of applying this approach to dynamic systems using the example of a classical problem in mechanics – the integrability of a rigid body, for which regular and quasi-regular orbits are localized and classified.

## 1 INTRODUCTION

The use of equations in graph-based semi-supervised learning (GSSL) enables a precise mathematical formulation of classification problems, offering rigorous justifications for algorithmic solutions. Moreover, the physical interpretations inherent in GSSL methods, drawing parallels with gas diffusion, Bernoulli processes, harmonic fields and waves, heat conduction, and other physical phenomena, provide intuitive and meaningful insights into the processes underlying label propagation. The most widely recognized GSSL methods employ energy functions in combination with the graph Laplacian. Prominent among these methods are Label Propagation, Label Spreading, p-Laplace Propagation, Deformable Laplacian, and Poisson Learning. A key advantage of these methods is their ability to efficiently propagate labels throughout the graph using a minimal amount of labeled data. Through iterative computational procedures, these methods maintain high accuracy.

Currently, there exist numerous methodologies for implementing semi-supervised learning techniques using the Laplacian operator. Comprehensive reviews and analyses of these methodologies have been presented in several influential works, including Zhu (2005), Calder et al. (2020), Song et al. (2022), Avrachenkov & Dreveton (2023), Aromal & Rasool (2021), Garcia-Cardona et al. (2013), Chapelle et al. (2009) and others. Among the most extensively studied and established Laplacian-based semi-supervised methods are Label Propagation, Label Spreading, and Poisson Learning. Each of these methods possesses distinct strengths and limitations. Specifically, Label Propagation and Label Spreading exhibit slow convergence rates on large datasets and tend to gradually lose memory of the initial labeled points through iterative updates. Recent advances have introduced a novel approach known as Poisson Label Propagation (also termed Poisson Learning). This method addresses several critical limitations of traditional Laplacian-based methods, notably their inefficiency in scenarios involving small quantities of labeled data. However, in other aspects, Poisson Learning shares similarities with the standard Label Propagation approach.

In our investigation, Poisson Learning was utilized to thoroughly assess its effectiveness for propagating labels. Previous studies have applied graph-analytical methodologies, particularly employing Poincaré sections along with their numerical implementations, to project trajectory structures from the phase spaces of dynamical systems onto two-dimensional manifolds. These projections yield two-dimensional data structures comprising distinct classes of periodic solutions (represented as closed curves). The trajectories are computed sequentially in real-time, consisting of discrete points derived from numerical solutions to systems of differential equations, typically solved using the Runge-Kutta method.

However, the practical implementation of this method faces significant limitations due to cumulative numerical errors, posing a major challenge for fully automated solution classification. Consequently, within a finite observation window, trajectory data obtained on two-dimensional Poincaré sections exhibit varying characteristics, necessitating accurate classification into distinct solution classes. Graph-based semi-supervised learning methods have emerged as advanced and effective tools for tackling a wide range of classification problems, accommodating scenarios involving two or more distinct classes. These machine learning approaches are particularly advantageous when fully supervised methods are impractical, provided the underlying structure of the data can be appropriately represented by a graph—a condition ideally suited to the context of our study.

In this article, we investigate the application of graph-based semi-supervised learning techniques for classifying dynamical system solutions based on two-dimensional trajectory data. To enhance classification accuracy and effectiveness, this research proposes employing a semi-supervised learning approach founded on nonlinear Poisson equations (Poisson Learning). Furthermore, we introduce a novel modification to Poisson Learning that integrates Nesterov optimization techniques, significantly improving the efficiency and accuracy of label propagation tasks.

## 2 COMPUTER ANALYSIS RELATED RESEARCH

Currently, a fairly not more large number of works are devoted to the study of the application of different machine learning methods for solving problems of locating periodic orbits of dynamical system. We will consider some of these works in more detail Das et al. (2012), Neimark et al. (2005), Petalas et al. (2008), Petalas et al. (2019).

In work Das et al. (2012) the detection of periodic orbits bears significance for the study of nonlinear mappings, since they can reveal crucial information on their dynamics. Recently, population–based stochastic optimization algorithms were introduced to address problems where traditional gradient–based approaches failed. The efficiency of these approaches in a applications, triggered further research towards the development of more efficient variants. This work presents the principal concepts of applying concurrent stochastic population–based approaches for the detection of periodic orbits, and also reports new results attained by the appli cation of Memetic Algorithms on well–known chaotic maps for periodic orbits with high period.

In Petalas et al. (2008) authors are proposed a new approach for the identification of the reso nances appearing in symplectic maps. In the proposed methodology, they make use of Evolutionary Algorithms which are population based search strategies used for global optimization. Authors have applied the proposed methodology to the 2-dimensional (2D) Henon map and obtained promising results which can be generalized to symplectic maps of higher (2m) dimensions. As is well-known, such maps are representative of Hamiltonian systems and occur in many physical applications.

In Petalas et al. (2019) authors presents an algorithm for automatic detection of round shapes on complex and noisy images. Algorithm based on a hybrid technology consisting of simulated annealing and differential evolu tion. New fuzzy target function was obtained on the boundary map of the input Images. Minimizing this function with the hybrid differential evolution firing algorithm results in automatic detection of circles in the image. Simulation of the results ends with several synthetic as well as natural images with variable complexity, which confirms the effectiveness of the proposed technique in terms of its ultimate accuracy, speed and reliability. In Kondratiev & Lyaptsev (2012) non-autonomous systems are studied by examining their Poincaré maps. Poincaré maps are used to identify periodic and subharmonic solutions and to study systems whose solutions exhibit chaotic behavior.

# 3 GRAPH-BASED SEMI-SUPERVISED POISSON LEARNING FOR TASKS CLASSIFICATION

## 3.1 PROBLEM STATEMENT

We consider the problem of classifying datasets from two classes represented on a on the trajectory flow are obtained on the two-dimensional Poincaré sections, which is discrete two-dimensional datasets. This datasets can has several classes. We will used a graph-based semi-supervised Poisson learning method for task classification this datasets. We used this method because we have small start label data in beginning.

So, we consider GSSL and first step is building a graph on the datasets. For the set of vertices $X = \{x_1, x_2, \ldots, x_n\}$ of some undirected graph $G$ and the first $m$ vertices (initial ones) of the set $X$ receive the corresponding labels $\{y_1, y_2, \ldots, y_m\}$. The number of labeled data is less than the total, i.e., $m < n$. The task of graph-based semi-supervised learning using a graph is to spread the label values from the initial vertices to all others, that is, to find the values labels for $\{x_{m+1}, x_{m+2}, \ldots, x_n\}$.

General Steps of the GSSL Method The key steps in the Graph-Based Semi-Supervised Learning (GSSL) method follow the framework outlined in Zhu (2005).

The initial step involves constructing a weighted graph by computing similarity measures between data points. Common techniques include geometric distances, Gaussian kernels, and K-Nearest Neighbor (KNN)-based distances. The selection of an appropriate weighting method depends significantly on the data's intrinsic distribution. Specifically, KNN-based weighting approaches generally yield superior performance when dealing with datasets characterized by overlapping or closely positioned classes Zhu (2005). A properly weighted graph ensures that the label propagation process maintains consistency with the underlying data structure, improving classification accuracy. Label initialization requires selecting a subset of representative data points from each class to serve as labeled references. To achieve balanced and unbiased initial labeling, it is advisable to allocate an equal number of labeled samples across all classes. In practice, different strategies can be used for label selection, such as random sampling, density-based sampling, or uncertainty-based selection. Proper initialization plays a critical role in ensuring that label propagation yields accurate and stable results. A penalty function is formulated based on the Dirichlet energy function defined over the graph. This function encourages smoothness in label propagation while preserving the information from the initially labeled nodes. Additional terms are incorporated to impose regularization constraints or boundary conditions, enhancing the stability and robustness of the solution. Common regularization methods such as L2 are applied to prevent overfitting, improve generalization capability, and ensure better performance on unseen data. Regularization plays a crucial role in mitigating the effects of noise and irregularities in the dataset. Minimizing the penalty function over the entire graph corresponds mathematically to solving the Poisson equation. In scenarios involving simplified energy considerations, this reduces to solving the Laplace equation. The numerical solution of these equations serves as the foundation for label propagation. In this research, the Nesterov optimization method is specifically employed for minimizing the penalty function. This advanced optimization technique accelerates convergence by incorporating momentum-based updates, enhancing computational efficiency, and significantly improving the accuracy and reliability of the label propagation process. The integration of Nesterov optimization ensures that the iterative updates of label values are more stable and converge faster compared to standard gradient-based methods. By following these structured steps, the GSSL framework enables effective label propagation and classification, even in scenarios where labeled data is sparse. This approach is particularly well-suited for applications involving complex data distributions, where graph-based methods provide a more flexible and scalable alternative to traditional supervised learning techniques.

## 3.2 GRAPH CONSTRUUCTION AND POISSON LEARNING

Let there be given an undirected weighted graph $G = G(X, V, W)$, where $n$ number of vertices. $X = \{x_1, x_2, ..., x_n\}$ - the set of vertices of the graph, $V$ - the set of edges, $W = (w_{ij})_{i,j=1}^{n}$ - is the weight matrix of the graph $G$.

The weight of the graph edges can be calculated using the following formulas (1):

$$w_{ij} = \psi\left(\frac{|x_i - x_j|}{\varepsilon_k\left(x_i\right)}\right) \text{ - } \textit{KNN weight, } \ w_{ij} = \exp\left(\frac{-|x_i - x_j|^2}{2\sigma^2}\right) \text{ - } \textit{Gaussian weight.} \qquad (1)$$

Where $\psi$ - some function, $\varepsilon$ - neighborhood, $\sigma$ is a parameter that controls the variance of neighbors. The degree of the node $x_i$ is determined by the formula $d_i = \sum_{j=1}^{n} w_{ij}$.

According to Poisson learning Calder et al. (2020), label propagation occurs by solving the Poisson equation, which has the form (2):

$$L\left(u_i\right) = \sum_{j=1}^{n} w_{i,j}\left[u\left(x_i\right) - u\left(x_j\right)\right] = \sum_{j=1}^{m}\left(y_j - \bar{y}\right)\delta_{ij}, \quad 1{\leq}i{\leq}m$$
$$L\left(u_i\right) = 0, \quad m+1{\leq}i{\leq}n \qquad (2)$$

where $L$ - is the non-normalized Laplace operator, $x_i$ - are the vertices of the undirected weighted graph, $y_i = y\left(x_i\right)$ - are the initial labels of the graph vertices, $u_i = u\left(x_i\right)$ - is the function of graph vertex labels $G$, $w_{ij}$ - edge weight $\left(x_i, x_j\right)$, $n$ - is the total number of graph vertices, the first $m$ of which are considered labeled. $\bar{y} = \frac{1}{m}\sum_{j=1}^{m} y_j$. $\delta_{i,j}$ - is the Kronecker symbol. $\sum_{i=1}^{n} d_i u\left(x_i\right) = 0$.

In an iterative process for solving this system based on the solution of the diffusion equation was proposed - Poisson Learning (Poisson Label Propagation) method. We note some advantages of Poisson Label Propagation over the classical Label Propagation (Laplace Label Propagation).

Key Advantages and Challenges of Poisson Label Propagation: 1) Increased Computational Efficiency: One of the main advantages of Poisson Label Propagation is its significantly higher computational efficiency. Unlike stochastic methods that rely on probabilistic updates, Poisson-based techniques utilize exact differential methods. This results in faster convergence rates and shorter execution times, making the approach well-suited for large-scale datasets.

2) Robustness Against Degeneracy in Low-Label Regimes: A persistent challenge in semi-supervised learning is the issue of degeneracy when the number of labeled vertices is very small relative to the total number of vertices. This occurs when the ratio of labeled to unlabeled vertices approaches zero, leading to unreliable label propagation. Poisson Learning overcomes this problem by leveraging the structure of the Poisson equation, ensuring that label information propagates effectively even in scenarios with extremely limited labeled data.

3) Addressing the "Forgetting" Problem in Large Datasets: Traditional Laplace-based label propagation methods suffer from a phenomenon known as "forgetting," where the initially assigned labels become diluted over iterative updates, especially in the presence of large unlabeled datasets. This issue arises when a small set of labeled vertices is surrounded by a vast number of unlabeled points, making it possible for their original labels to be reclassified erroneously. Poisson Learning mitigates this effect by incorporating a penalty function that retains initial label constraints while ensuring effective propagation.

4) Enhanced Convergence Speed for Large-Scale Data: A crucial limitation of many graph-based learning methods is their slow convergence when applied to large-scale datasets. To address this, various optimization techniques have been proposed to accelerate convergence. Among these, the Nesterov optimization method has been identified as one of the most effective approaches. By incorporating momentum-based updates, Nesterov optimization significantly improves convergence speed, allowing Poisson Learning to achieve stable and accurate classification results more efficiently.

### 3.3 POISSON LEARNING VIA NESTEROV OPTIMIZATION

Let's consider the problem of semi-supervised learning using Poisson equations more thoroughly. Poisson learning also has a variational interpretation, which reduces semi-supervised learning to minimizing the Dirichlet energy. Dirichlet energy is often used as a penalty function. The Dirichlet energy formula can be represented as (3)

$$E(u) = \sum_{i,j=1}^{n} w_{ij} \left| u(x_i) - u(x_j) \right|^2 - \sum_{j=1}^{m} (y_j - \bar{y}) u(x_j) \tag{3}$$

In this work, to find the minimum, the Nesterov algorithm is employed, which demonstrates better results compared to gradient descent.

For the Nesterov approach, the optimization step looks like this

$$g_{\{k\}}(x_i) = \frac{1}{d_i} \left( \sum_{j=1}^{m} (y_j - \bar{y}) \delta_{ij} - \sum_{j=1}^{n} w_{ij} \left( u_{\{k\}}(x_i) - u_{\{k\}}(x_j) \right) \right) \tag{4}$$

$$v_{\{k\}} = \gamma v_{\{k-1\}} - \alpha g_{\{k\}}, \quad u_{\{k+1\}}(x_i) = u_{\{k+1\}}(x_i) + v_{\{k\}}.$$

Where $g_{\{k\}}$ - is the gradient at any time step k, $\alpha$ - hyperparametr.

The label selection rule can be written as (5):

$$u_l(x_i) = argmax_{j \in 1,2} s_j u_j(x) \tag{5}$$

where $s_j = \left( \frac{b_j}{\underline{y}} \right)$, $b_j$ - is the share of data belonging to the class $j \in 1, 2$.

Let's consider a graph-based semi-supervised learning method based on Poisson learning and Nesterov optimizer. This algorithm will consist of the following steps.

---

**Algorithm 1** Algorithm PoissonNesterov

---

1: **procedure** PNFRAMEWORK($W, y, T$)
2:     Initialization $F = [y_1, y_2, \ldots, y_m]$
3:     Compute Degree Matrix: $D = \mathrm{diag}(W\mathbf{1})$
4:     Compute Laplacian: $L = D - W$
5:     Compute Average Label Vector: $c = \frac{1}{m} F\mathbf{1}$
6:     Construct Matrix: $B = [F - c, \mathrm{Z}(2, n - m)]$
7:     Initialize: $U = \mathrm{Z}(n, 2)$
8:     **for** $i = 1$ to $T$ **do**
9:         Update $U_j^k$ using the update formula (4) - (5)
10:    **end for**
11:    Label Assignment: $l_k = \arg\max_{1 \le j \le 2} U_j^k$
12:    **return** $u_l = [l_1, l_2, \ldots, l_n]$
13: **end procedure**

---

### 3.4 COMPARISON AND ANALYSES

We will compare with existing methods and their modifications: Label Propagation, Label Spreading, Poisson GradientDescent, Poisson Nesterov on classical model "Two Moons" with 10000 datasets and $\{1,2,3,4,5,6,7,8,9,10\}$ numbered label per class.

The picture illustrates how the accuracy of four semi-supervised learning methods (Label Propagation, Label Spreading, Poisson GradientDescent and Poisson Nesterov) changes depending on the number of labeled instances per class (shown on the X-axis). The Y-axis represents the final accuracy in percentage.

Overall trend: as the number of labels per class increases, all methods improve in accuracy. Highest accuracy is achieved by Poisson Nesterov (brown curve): starting at about 80% with just one label per class, it reaches around 92% at ten labels. Second place goes to Poisson GradientDescent (red curve), which rises to about 90% at ten labels.

Label Propagation (blue curve) and Label Spreading (orange curve) yield lower results compared to the Poisson-based methods. By ten labels, Label Propagation approaches about 55%, while Label

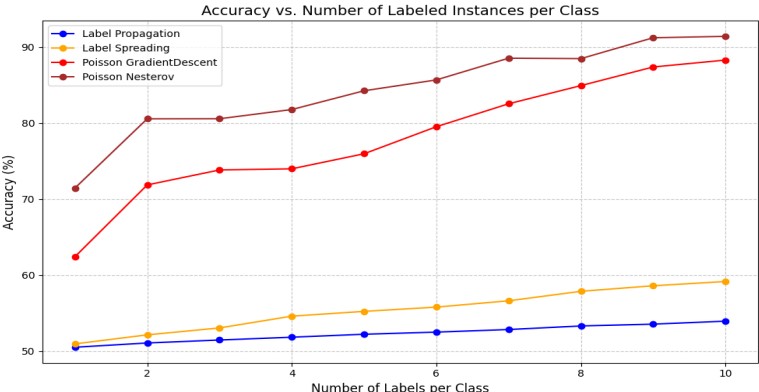

Figure 1: Comparison with other methods. Label Propagation, Label Spreading, Poisson Gradien-Descent and Poisson Nesterov.

Spreading reaches around 55–60%. In summary, as the number of labeled samples increases, all algorithms become more accurate. However, the Poisson Learning–based methods significantly outperform Label Propagation and Label Spreading, especially with a sufficient number of labels.

With this approach and modification, the task of classifying the periodic solutions of the dynamical system is solved with sufficient accuracy even in real-time mode, taking into account accumulation errors and time constraints.

## 4 APPLICATION OF POISSON LEARNING FOR THE CLASSIFICATION OF SOLUTIONS OF DYNAMICAL SYSTEMS

### 4.1 GENERAL CONCEPTION OF CLASSIFICATION OF REGULARITY SOLUTIONS

The existing methods of KAM theory for studying dynamical systems provide a clear characterization of a system and predict its behavior under given initial conditions, whether regular, quasi-regular, or chaotic. However, performing a comprehensive analytical study of dynamical systems is challenging in practice due to their high dimensional and numerous parameters. Consequently, the solutions obtained are often highly localized and approximate. When numerical and computational methods are employed, errors and cumulative numerical inaccuracies arise, making the investigation of global regular cases particularly difficult Neimark et al. (2005), Petalas et al. (2008), Ruchkin (2014),Kondratiev & Lyaptsev (2012).

A useful approach for numerical studies of phase space, which consists of a set of non-intersecting phase trajectories, is through Poincaré sections. These sections, constructed in phase space, reduce the dimensional of the system by one. Dynamical systems of third and fourth order are of particular interest, as their Poincaré sections produce graphical representations either on a plane or in three-dimensional space. When phase flow points form a curve, the system exhibits regular behavior, either periodic or multi-periodic, as observed in Hamiltonian systems.

A major research focus is the reconstruction and analysis of global Poincaré sections, which provide insights into all possible motions of the system. Although Poincaré sections are approximated using numerical integration over a fixed time interval, they still offer valuable insights into the overall behavior of Hamiltonian systems. The resulting phase portraits of two- and three-dimensional Poincaré sections can be analyzed using statistical or deterministic pattern recognition techniques .

In modern studies, the significance of new results in dynamical systems often relies on computational analysis and advanced numerical techniques. New and particular interest approach are machine learning methods - Label Propagation for ask classification solve of dynamical system. It represents a new technique for studying dynamical systems and holds great promise for the automatic detection of regular and chaotic behaviors in such systems with spatial program Avdyushina et al. (2023).

## 4.2 PROBLEM STATEMENT FOR HAMILTONIAN SYSTEM

Let a dynamical system of Hamiltonian type of even order be given, for example two. Assume that the system possesses an energy integral, i.e., the Hamiltonian is defined as $H(q,p), (q,p) \in \Omega \subset \mathbb{R}^{2n}$, and the system is described by Hamilton's equations:

$$\dot{q} = \frac{\partial H(q,p)}{\partial p}, \quad \dot{p} = -\frac{\partial H(q,p)}{\partial q}.$$

Assume that the system admits a global or local Poincaré section $S$. Choose a smooth section $S$ in the phase space defined by $g(q,p) = 0, \nabla g(q,p) \neq 0$.

For each initial condition $(q(0), p(0))$, consider the phase trajectory $\gamma(t) = (q(t), p(t))$ and record its intersections with $S$. As a result, we obtain a set of points $\mathcal{X} = \{x_i\}_{i=1}^{N} \subset S \subset \mathbb{R}^2$.

Assume that the trajectories are divided into $K$ classes corresponding to different dynamical regimes. For a subset of indices $L \subset \{1, 2, \ldots, N\}$, labels are provided: $y_i \in \{1, 2, \ldots, K\}, \quad i \in L$. It is assumed that if the points $x_i$ and $x_j$ (corresponding to close initial conditions) are nearby, then their trajectories belong to the same class.

The set of points forms a trajectory structure of the phase space, which in the case of numerical integration is discrete and can be divided into classes using machine learning methods.

The goal article is to construct a classifier $f: S \to \{1, 2, \ldots, K\}$, which assigns to each unlabeled point $x_i$, for $i \in U = \{1, \ldots, N\} \setminus L$, a label $\hat{y}_i = f(x_i)$. To this end, we employ graph-based Semi-Supervised Learning method based on a graph model.

Thus, using the Hamiltonian equations, the construction of a global (or local) Poincaré section, and a graph-based model, we obtain a rigorous mathematical formulation of the semi-supervised classification problem for the trajectory structure of the phase space using the Poisson Label Propagation method.

## 4.3 REVIEW OF RELEVANT LITERATURE OF POINCARÉ SECTIONS

Let's consider the works devoted to the construction of the analysis of Poincaré sections for various physical problems Zhang et al. (2019), Lerma-Hernández et al. (2018), Liu et al. (2024), Kovaleva et al. (2018), Caracciolo et al. (2024).

Zhang et al. Zhang et al. (2019) explore the stabilization of tearing modes in plasmas using modulated electron cyclotron current drive. Their study provides insights into the control of plasma instabilities in fusion devices and demonstrates the impact of current modulation on stability.

Lerma-Hernández et al. Lerma-Hernández et al. (2018) present an analytical description of the survival probability of coherent states within regular regimes. This work contributes to the understanding of dynamical evolution in quantum systems by characterizing long-term behavior under different regimes.

Liu et al. Liu et al. (2024) propose a method for discriminating tokamak sawtooth crash models using localized density and temperature measurements. Their findings enhance the capability to differentiate between competing theoretical models based on experimental diagnostics.

Kovaleva et al. Kovaleva et al. (2018) analyze the forced pendulum within the framework of engineering mechanics. Their study highlights the complex behavior of forced oscillatory systems, emphasizing deterministic chaos and stability conditions.

Caracciolo et al. Caracciolo et al. (2024) investigate the three-dimensional orbital architecture of exoplanetary systems using KAM stability analysis. Their research contributes to celestial mechanics by assessing long-term stability criteria for exoplanets.

The conducted analysis of the considered and other works shows that at present there are no known works devoted to the application of machine learning methods for classifying solutions of dynamic systems on Poincaré sections in the plane and space. This direction of research, as well as the obtained results, is new.

### 4.4 EXAMPLE OF MECHANICAL SYSTEM

The problem of the motion of a rigid body with a fixed point is considered in the classical framework, addressing the direct problems of mechanics. Given the static, kinematic, and dynamic (structural) parameters of a rigid body with a fixed point and the initial conditions of motion, it is necessary to determine its trajectory in space at any given moment, classify the type of motion, and establish the nature of the body's dynamics.

A mathematical model describing the motion of a free rigid body in a mobile reference frame is given by a system of six ordinary differential Euler equations:

$$J\dot{\omega} = J\omega \times \omega + r \times \nu, \quad \dot{\nu} = \nu \times \omega, \tag{6}$$

where $J = \mathrm{Diag}(A, B, C)$ represents the inertia tensor, $\omega = (p, q, r)$ is the angular velocity of the body in the mobile frame, $\nu = (\nu_1, \nu_2, \nu_3)$ is the unit vertical vector, and $r = (r_1, r_2, r_3)$ is the vector from the fixed point to the center of mass of the body.

Equation (6) determines the phase space $R^{12}$ as $R^6(\omega, \nu) \times R^6(J, r)$, defining the family of possible (both regular and chaotic) trajectories of the dynamical system. The initial conditions are given by $\omega_0 = \omega(0)$ and $\nu_0 = \nu(0)$.

In the regular case, the dynamical system exhibits "well-behaved" trajectories that remain stable under small perturbations of initial conditions, allowing long-term predictability. In contrast, chaotic motion is characterized by an extreme sensitivity to initial conditions, leading to exponentially unstable trajectories. The chaotic case limits predictability to a finite time interval, known as the time horizon.

The aim of previously article is to develop an interactive computational system to solve system (6), classify its behavior system. If system (6) can be integrated explicitly, it is possible to distinguish between regular and chaotic cases.

#### 4.4.1 INTEGRABILITY AND NUMERICAL SOLUTION

It is known in previously, that the solution of system (6) can be reduced to quadratures if four integrals are found. However, for arbitrary values of the system parameters, only three first integrals exist:

$$H(\omega, \nu) = \frac{1}{2}J\omega \cdot \omega - r \times \nu = h, \quad G(\omega, \nu) = J\omega \cdot \nu = g, \quad I(\nu, \nu) = \nu \cdot \nu = 1. \tag{7}$$

A fourth integral exists only in three well-known cases: Euler-Poinsot, Lagrange-Poisson, and Kowalewski, as well as in some special cases. These cases are characterized by specific parameter sets, allowing the reduction of system (6) to a system of algebraic equations. In integrable cases, the phase space trajectories are deterministic and exhibit stable motion. For non-integrable cases, the system's trajectory cannot be expressed analytically, necessitating numerical integration using the fifth-order Runge-Kutta method. The nature of the motion is determined by analyzing phase-space trajectories. For a fixed point in $R^6(J, r)$, the first integrals (7) define a compact three-dimensional invariant manifold $Q_{h,g}^3$ in $R^6(\omega, \nu)$, which governs the system's phase flow. The spatial organization of $Q_{h,g}^3$ within $R^3(\omega)$ and $R^3(\nu)$ provides insight into the evolution of physical quantities described by equation (6).

#### 4.4.2 QUALITATIVE ANALYSIS AND COMPUTATIONAL APPROACH

The qualitative analysis of phase space structures relies on Poincaré sections and the theory of invariant curves. The onset of chaotic behavior is identified when the Poincaré section appears as a scattered cloud of points, forming a two-dimensional region. In contrast, regular motion corresponds to phase trajectories confined to smooth curves.

To study the topology of $Q_{h,g}^3$, integrable cases are analyzed using global sections $P^2 \subset Q_{h,g}^3$. For non-integrable cases, these sections are represented on the Poisson sphere $S^2$ with $\|\nu\| = 1$. The

global section $P^2$ forms a compact two-dimensional manifold within $Q^3_{h,g}$, with phase trajectories either entirely filling $Q^3_{h,g}$ or intersecting $P^2$.

A computational program was developed for the intelligent analysis of regular and chaotic dynamics in mechanical systems, demonstrating the feasibility of this approach Ruchkin (2014). An example of recognizing regular and chaotic motion using a three-dimensional Poincaré section is illustrated in Figure. This Poincaré section was computed via the Runge-Kutta 4-5 method for system (6) and reconstructed on the Poisson sphere using equation (7).

### 4.4.3 RESULT OF CLASSIFICATION

So, in the integrable cases Poincare section will be of a plane (surface area), which shows a certain type of closed curves: circles, ellipses, set of circle and more. These cases correspond to the regular (quasi-periodic) solution of a nonlinear dynamical system. We use Machine Learning for classification this cases. With help computer program we labeled start points from this classes ("Yellow points"). "White dots" refers to the classified trajectory. "Red dots" refers to the not classified set of points. (Fig.2 - Fig.5.) These figures show examples of curve extraction and classification in four most typical cases: regular orbit, multi-regular case, multi-regular case 2.

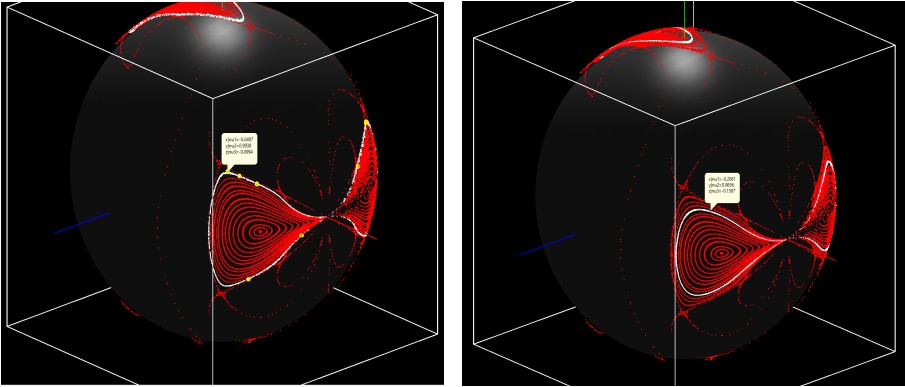

Figure 2: Example of classification of regular orbits.

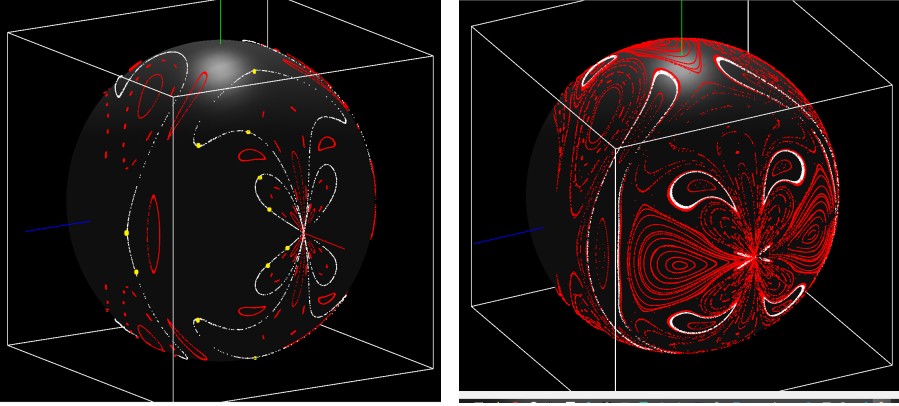

Figure 3: Example of classification of multi - regular orbits in case 1.

## 5 CONCLUSION

In this paper we have consider the problem of classification periodic and multi-periodic orbits of dynamical systems using graph-based semi-supervised Poisson Learning method. The detection of periodic orbits of dynamical systems has carried out by means of an analysis of the Poincare' sections of phase space on the plane or 2d - datasets.

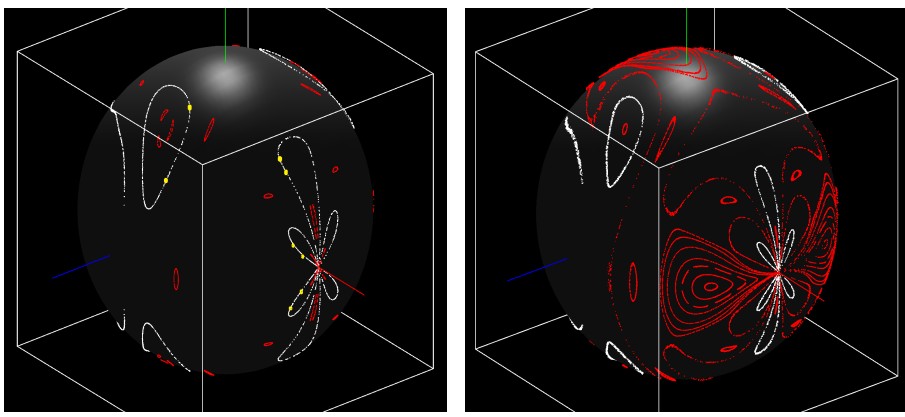

Figure 4: Example of classification of multi - regular orbits in case 2.

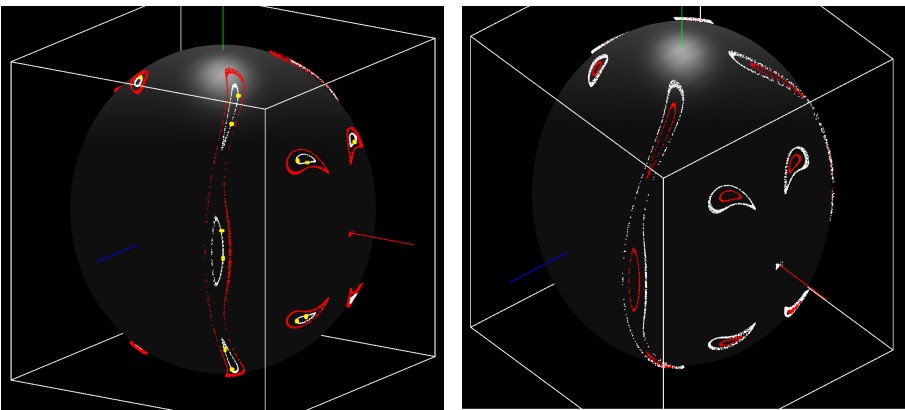

Figure 5: Example of classification of resonance orbits.

In work a modification of Graph-based semi-supervised learning via Poisson equation is proposed. Furthermore, to find the minimum loss function, the Nesterov algorithm is employed, which demonstrates better results compared to gradient descent. With this approach and modification, the task of classifying the periodic solutions of the dynamical system is solved with sufficient accuracy even in real-time mode, taking into account accumulation errors and time constraints.

Poisson learning offers advantages over traditional Laplace-based approaches, requiring significantly fewer labeled samples to achieve the desired classification accuracy while also reducing computational time.

To enhance convergence, we propose a modification of Poisson learning by incorporating the Nesterov algorithm for optimizing the loss function. This approach outperforms classical gradient descent and achieves 92% accuracy with as few as 10 labels per class, even when accounting for accumulation errors and time constraints.

The proposed approach for investigation dynamic systems expands the possibilities of analytical and numerical use of KAM theory and is the next step in building a computer system for conducting a complete automatic investigation of dynamic systems.

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
