# OpenReview forum: "Application of Semi-Supervised Graph-Based Learning for the Classification of Solutions of Dynamical Systems"
_mathai.club/MathAI/2025/Conference — MathAI 2025 Oral_

### Official Review · Reviewer_vcmo · 2025-02-27
**A careless article with unsubstantiated conclusions(updated)**

**Rating:** 7
**Confidence:** 4

**Review:**

The article presents graph-based semi-supervised learning method using Nesterov's algorithm to solve the classification problem of regular orbits in the Poincaré problem.

**Strengths:**
- A detailed algorithm for solving the problem set in the article is presented.
- The Poisson learning method is explained.
- A visual comparison of the proposed method with the exact solution is provided.
- Detailed description of the mathematical problem being addressed, existing challenges, and an outline of the numerical and integral solutions provided.

**Weaknesses:**
- The impact of Poisson learning modification is unclear, as there is no comparison available.
- The references do not follow the required format.

---

### Official Review · Reviewer_k5r3 · 2025-02-27
**Application of Poisson Learning for the Classification of Solutions of Dynamical Systems**

**Rating:** 8
**Confidence:** 5

**Review:**

The paper applies the Poisson learning method to graph neural networks using Nesterov's algorithm for classifying regular orbits in the Poincare problem. Strengths are the next: detailed algorithm presentation, visual comparison with the exact solution. However, providing more comprehensive details about the experimental setup would greatly enhance the clarity and understanding of the methodology employed.

---

### Decision · Program_Chairs · 2025-03-08

**Decision:**

Accept (Oral)

**Comment:**

Your article has been accepted and you can make a presentation on the article. All articles will be sorted by rating and within the available conference places one author from each article will be invited. If there are not enough places, then you will either have the opportunity to present remotely or come at your own expense!